# Precipitation Dominates the Distribution of Species Richness on the Kunlun–Pamir Plateau

Xiaoran Huang [1,2,3,4], Anming Bao [2,5,6,*], Junfeng Zhang [7], Tao Yu [2,3,4], Guoxiong Zheng [8], Ye Yuan [2,3,4], Ting Wang [2,3,4], Vincent Nzabarinda [2,3], Philippe De Maeyer [4,9] and Tim Van de Voorde [4,9]

[1]  Key Laboratory of Smart City and Environment Modelling of Higher Education Institute, College of Resources and Environment Sciences, Xinjiang University, Urumqi 830046, China
[2]  State Key Laboratory of Desert and Oasis Ecology, Xinjiang Institute of Ecology and Geography, Chinese Academy of Sciences, Urumqi 830011, China
[3]  University of Chinese Academy of Sciences, Beijing 100049, China
[4]  Department of Geography, Ghent University, 9000 Ghent, Belgium
[5]  Sino-Belgian Joint Laboratory of Geo-Information, Urumqi 830011, China
[6]  China-Pakistan Joint Research Centre on Earth Sciences, CAS-HEC, Islamabad 45320, Pakistan
[7]  Research Centre for Ecology and Environment of Central Asia, Chinese Academy of Sciences, Urumqi 830011, China
[8]  College of Earth and Environmental Sciences, Lanzhou University, Lanzhou 730000, China
[9]  Sino-Belgian Joint Laboratory of Geo-Information, 9000 Ghent, Belgium
*  Correspondence: baoam@ms.xjb.ac.cn

**Abstract:** The Kunlun–Pamir Plateau is a globally irreplaceable biodiversity reserve, yet it is still unclear what causes the distribution of species richness. Here, we relied on the productivity and the water–energy dynamics hypotheses to investigate the distribution pattern of species richness (and its determinants) in the Kunlun–Pamir Plateau. The productivity hypothesis is mainly based on five MODIS products (NDVI, EVI, FPAR, LAI and GPP), which were calculated for three Dynamic Habitat Indices (DHIs): (1) cumulative productivity (CumDHI), (2) minimum productivity (MinDHI) and (3) intra-annual variation productivity (VarDHI). The CumDHI was applied to assess whether or not more energy has a higher species richness value. The MinDHI was used to determine and evaluate the higher minimums, leading to a higher species richness. The VarDHI was the annual variation value in productivity and was utilized to assess if the reduced intra-annual variability triggers a higher species richness. We found that the DHIs based on the FPAR correlated slightly higher with the mammal, bird, breeding bird and non-breeding bird richness (than those based on the other four DHIs, and the values were 0.24, 0.25, 0.24 and 0.01, respectively). The correlation between the climate variables and the mammals, birds, breeding birds and non-breeding birds was bigger at 0.24, 0.54, 0.54 and 0.02, respectively, and was mainly dominated by the precipitation-related climatic factors. The water–energy dynamic hypothesis is better suited to the Kunlun–Pamir Plateau than the productivity hypothesis. Our results might provide valuable information regarding the biodiversity conservation in this region.

**Keywords:** species richness; dynamic habitat index; productivity hypothesis; water–energy dynamic hypothesis; Kunlun–Pamir Plateau

## 1. Introduction

The high value species richness means a rich regional biodiversity. The research of species richness is the basis for constructing evolutionary, ecological models and conservation strategies [1,2]. The global species richness has decreased during the last decades [3], and this trend will continue [4]. Several theoretical hypotheses, such as the productivity hypothesis [5,6], water–energy dynamic hypothesis [7,8], ambient energy hypothesis [9,10], cold tolerance hypothesis [8,11] and the metabolic theory of ecology [12–14], were used to explain the distribution patterns of species richness. Among those hypotheses, the

productivity hypothesis and the water–energy dynamic hypothesis were the most widely used. The productivity hypothesis suggests that the higher the productivity, the higher the species richness [5,6,15,16]. The main idea of the water–energy dynamics hypothesis (sometimes referred to as the moisture hypothesis) showed that the large-scale patterns of species richness are determined by a combination of water and energy [7]. The energy in this hypothesis is usually expressed by the potential evapotranspiration, temperature or moisture. The water is generally referred to as liquid water and is commonly expressed as the annual rainfall of a region [7,17]. Some studies have applied climate data to present energy and moisture variables [18,19].

Satellite observations provide a rich, long-time series of vegetation productivity products. They can be used to test the productivity hypothesis [20,21]. The Moderate Resolution Imaging Spectroradiometer (MODIS) data collected by NASA's Terra and Aqua satellites offered various products. This could be utilized for species richness assessments at different scales and include the vegetation indices, such as the Normalized Difference Vegetation Index (NDVI) and Enhanced Vegetation Index (EVI)), Leaf Area Index (LAI) and Fraction of absorbed Photosynthetically Active Radiation (FPAR), as well as Gross Primary Productivity (GPP). The most suitable productivity products vary from region to region. The NDVI and EVI belong to the same MODIS vegetation dataset. The NDVI is the best indicator of vegetation growth status and vegetation cover; however, overfitting exists in densely vegetated areas due to nonlinear transformations. The EVI improves on NDVI saturation. The FPAR and LAI are based on reflectance values of up to seven MODIS spectral bands, while the NDVI and EVI are based on two and three bands, respectively, so the FPAR and LAI provide a closer proxy for vegetation productivity than the NDVI and EVI. GPP products are mainly calculated using FPAR, light and effective radiation (PAR), as well as other surface meteorological data from remote sensing inversion using a light energy utilization model, which is good information for monitoring vegetation growth. Five MODIS products allow a multifaceted and more comprehensive analysis of their relationship with species richness [22]. The Dynamic Habitat Indices (DHIs) based on the productivity hypothesis theory can be calculated from the MODIS products to demonstrate the distribution of species richness [23,24]. It has been proven that these correlate well with the species richness [22,25,26]. The DHIs comprise three indicators: the cumulative annual productivity (CumDHI), minimum productivity (MinDHI) and productivity variation (VarDHI). A good correlation between these indices and the species richness was confirmed on the local and global scales [22,26–32]. For example, the biodiversity in China has been detected by the FPAR-derived DHIs [31]. In the United States, the bird species richness correlated with the FPAR-derived DHIs on grasslands, while it denoted a better correlation with the LAI-derived DHIs on woodlands [22]. In Russia, FPAR-derived DHIs combined with climatic variables seemed promising to predict the abundance of large ungulates such as moose [32]. On a global scale, the multiple regression used GPP-derived DHIs that also adequately described the distribution of species richness [26]. The species richness was simultaneously impacted by a combination of various factors and processes but varied significantly across the scales and regions.

The Kunlun–Pamir Plateau is adjacent to the Tibetan Plateau, which constitutes the second-highest plateau in the world. It is located in the arid zone and is also an irreplaceable global biodiversity reserve. Its substratum is a complex, sensitive and vulnerable area for the global climate and environmental change [33]. Additionally, it has a continental alpine environment with long winters, a scarce precipitation, mainly bare land and desert grasslands. However, there is still a lack of research on the spatial distribution of species richness and their causes at this plateau.

Our goal was to use the productivity hypothesis based on DHIs and the water–energy dynamic hypothesis based on bioclimatic factors to explain the species richness distribution of the Kunlun–Pamir Plateau. Specifically, we sought (1) to derive and describe the DHIs at a 500 m resolution from the MODIS vegetation products on the Kunlun–Pamir Plateau and (2) to analyze the applicability of these two hypotheses in the area.

## 2. Data and Methods

### 2.1. Data

#### 2.1.1. MODIS Data

The MODIS products used to calculate the DHIs included the NDVI, EVI, FPAR, LAI and GPP in both 8- and 16-day composites and at a 500 m resolution (Table 1). All products from 2001 to 2020 were downloaded from the Application for Extracting and Exploring the Analysis Ready Samples (AEEARS) https://appeears.earthdatacloud.nasa.gov/ (accessed on 14 July 2022). These are all related to the vegetation productivity and could be used to validate the application of the productivity hypothesis [22,25,26].

**Table 1.** MODIS products used to calculate the DHIs.

| Index | Name | Platform | Temporal Resolution (Day) | Spatial Resolution (m) |
|---|---|---|---|---|
| NDVI | MOD13A1 | Terra | 16 | 500 |
| EVI | MOD13A1 | Terra | 16 | 500 |
| FPAR | MOD15A2H | Terra | 8 | 500 |
| LAI | MOD15A2H | Terra | 8 | 500 |
| GPP | MOD17A2HGF | Terra | 8 | 500 |

#### 2.1.2. Species Richness

The species richness data were downloaded from the website BiodiversityMapping.org (https://biodiversitymapping.org/index.php/download/ (accessed on 7 August 2022)) [34–36]. They are also from the IUCN [37] (http://www.iucnredlist.org (accessed on 7 August 2022)) and Birdlife International (http://datazone.birdlife.org/species/requestdis (accessed on 7 August 2022)) [38]. The maps currently comprise mammals, birds, breeding birds and non-breeding birds. Its spatial resolution is $10 \times 10$ km with the Eckert IV equal-area projection. Maps represent native extant species only.

#### 2.1.3. Climatic Data

The climatic data are used from Worldclim version 2.1 with a spatial resolution of $30''$ and include 19 bioclimatic variables (https://www.Worldclim.org/data/Worldclim21.html (accessed on 24 May 2022)) for 1970–2000. It was released in January 2020. The first eleven variables (BIO1–11) were applied to characterize the energy, and the last eight ones (BIO12–19) were selected to describe the water factors. The variation in vegetation was also impacted by the climate variables (Table 2).

**Table 2.** Nineteen Worldclim Bioclim variables used to test the water–energy dynamics hypothesis.

| | Variables | Description |
|---|---|---|
| Energy factor | BIO1 | Annual Mean Temperature |
| | BIO2 | Mean Diurnal Range |
| | BIO3 | Isothermality |
| | BIO4 | Temperature Seasonality |
| | BIO5 | Max Temperature of Warmest Month |
| | BIO6 | Min Temperature of Coldest Month |
| | BIO7 | Temperature Annual Range |
| | BIO8 | Mean Temperature of Wettest Quarter |
| | BIO9 | Mean Temperature of Driest Quarter |
| | BIO10 | Mean Temperature of Warmest Quarter |
| | BIO11 | Mean Temperature of Coldest Quarter |
| Water factors | BIO12 | Annual Precipitation |
| | BIO13 | Precipitation of Wettest Month |
| | BIO14 | Precipitation of Driest Month |
| | BIO15 | Precipitation Seasonality |
| | BIO16 | Precipitation of Wettest Quarter |
| | BIO17 | Precipitation of Driest Quarter |
| | BIO18 | Precipitation of Warmest Quarter |
| | BIO19 | Precipitation of Coldest Quarter |

*2.2. Methods*

2.2.1. Calculation of the DHIs

First, the 8-day and 16-day NDVI, EVI, FPAR and LAI products were synthesized into monthly data using the maximum synthesis method [22,26,39]. Then, the CumDHI, MinDHI and VarDHI were calculated for these indices from 2001 to 2021.

The CumDHI is the sum of the production values for one year. Since it signifies the number of annually available resources that could provide animals with resources such as food, it might indirectly indicate the species abundance [40]. Here, the calculation of the DHI for FPAR is applied as an example, and its equation is shown below:

$$\text{CumDHI} = \sum\nolimits_{\text{month}} \text{MAX}_{\text{layer, FPAR}} \tag{1}$$

where the layer refers to the monthly data.

The MinDHI stands for the minimum extent of vegetation cover in one year and represents the minimum capacity within one year. It could denote the minimum accommodation of food and habitat resources during a year [40]. It is calculated by means of the following equation:

$$\text{MinDHI} = \text{MIN} \left\{ \left( \text{MAX}_{\text{layer,Fpar}} \right)_{\text{month...}} \right\} \tag{2}$$

where VarDHI stands for the variation of the production values in one year. It refers to the natural resources associated with the habitat quality, such as food, water, nutrient substances, etc. within the year, and it might reveal the characteristics of the organisms' activity trajectory [28]. The following equation calculates the VarDHI:

$$\text{VarDHI} = \text{STD} \left\{ \left( \text{MAX}_{\text{layer,FPAR}} \right)_{\text{month...}} \right\} \tag{3}$$

where $\text{MAX}_{\text{layer,FPAR}}$ refers to the maximum FPAR value for one month, and the month indicates the 12 months of the year. MIN and STD represent the minimum and standard deviation of the maximum monthly value of FPAR for one year. The above-mentioned equations were applied to the NDVI, EVI, FPAR and LAI. The GPP was computed by replacing the maximum values with the cumulative values in the three equations.

2.2.2. Statistical Analysis

The Spearman correlation was utilized to quantify the degree of interchangeability between the same MODIS products-based DHI and different MODIS products. In order to verify whether or not climate variables impacted the DHIs (CumDHI, MinDHI and VarDHI), we performed a correlation analysis of the DHIs with 19 Bioclim variables. All correlation analyses were based on a random sample of 10,000 of the 1 km DHIs and bioclimatic variable pixels with the "no data" set to zero [22,26].

We computed the Spearman rank correlations of species richness, DHIs and Bioclim variables, and scatter plots were produced for visualization purposes. Multiple linear regression models estimated the dependent variables using the most available combinations of various independent variables to make the results more trustworthy. We conducted multiple linear regression analyses to predict the species richness based on two hypotheses, and the relative contribution of each model was conducted by hierarchical partitioning analyses using the R package hier.part [41]. The analysis was based on a sample of 10,000 of the 10 km resampled DHIs and Bioclim variables. The root mean square error (RMSE) and the adjusted coefficient of determination ($R^2$ adj) were used to evaluate the overall predictive power of a model under 95% confidence intervals.

## 3. Results

### 3.1. Distribution Patterns of DHIs

In the Kunlun–Pamir Plateau region, the spatial changes in the distribution of the cumulative DHI, minimum DHI and variation DHI values are rather insignificant. However, various MODIS products demonstrated different distribution patterns (Figure 1a–f). The DHIs of the same MODIS vegetation dataset indicated positive correlations. The correlation between the cumulative DHIs and the minimum DHI was moderately intense, except in the case of GPP, in which the correlation between the cumulative DHI and minimum DHI was rather small (0.23). The Spearman rank correlation ranged from 0.50 to 1. The correlations of variation DHIs with a cumulative DHI and minimal DHI were 0.43 to 1 and 0.21 to 0.81, respectively. The correlation between the GPP variation DHI and minimal DHI was 0.21 (Figure 2).

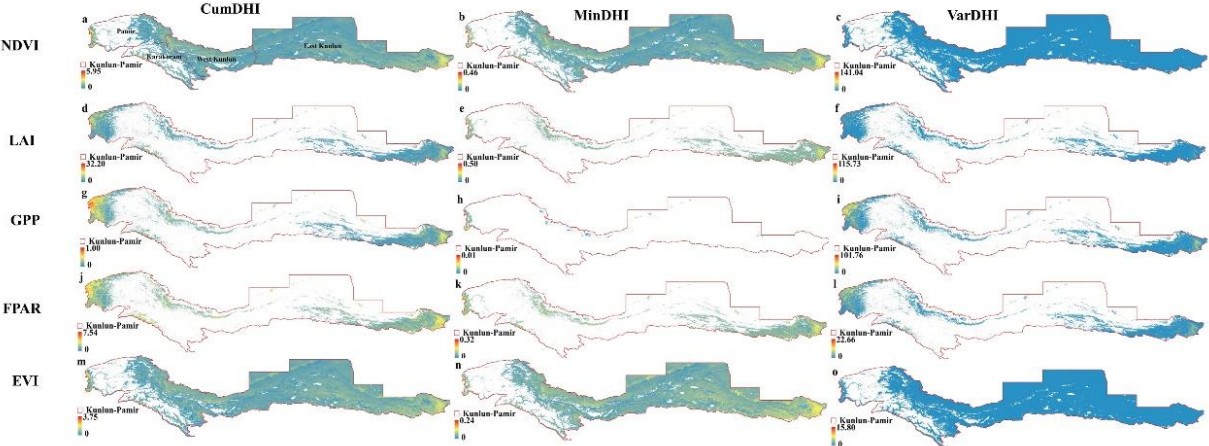

**Figure 1.** The spatial patterns of three DHIs derived from (**a**–**c**) the NDVI, (**d**–**f**) LAI, (**g**–**i**) GPP, (**j**–**l**) FPAR and (**m**–**o**) EVI.

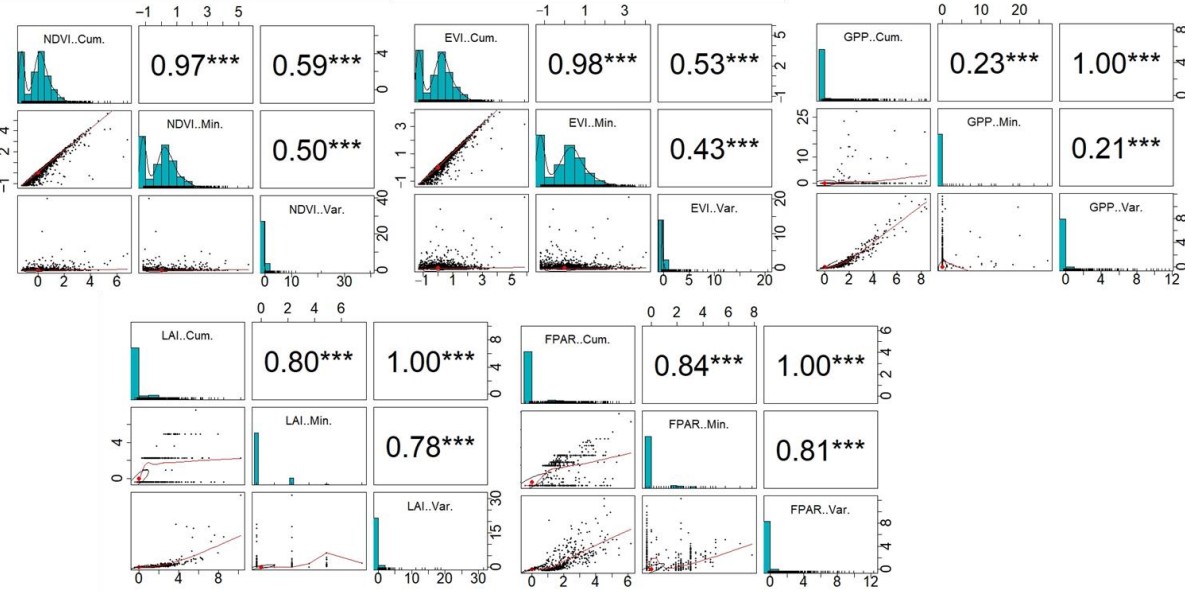

**Figure 2.** Spearman rank correlations and scatter plots among the three DHIs derived from the five MODIS products. It represents a correlation matrix for cum–min–var with scatter plots on one side of the diagonal and correlations on the other side. Significant relations: *** $p < 0.001$.

The DHI correlations from different vegetation datasets were uneven. Some of correlations were also nonlinear (Figure 2). The Spearman rank correlations of the same

vegetation products (e.g., NDVI- and EVI-based DHIs) were higher than the ones between the DHIs based on different vegetation products (e.g., FPAR- and GPP-based DHIs). The highest correlations values were noticed between the FPAR- and LAI-based DHIs (from 0.78 for the minimum DHI to 1 for the cumulative and variation DHI). The correlations between the cumulative and variation DHIs based on the GPP, FPAR and LAI were also substantial (Figures 2 and 3). The GPP-based DHI correlated the least with the DHI based on other vegetation products, except the cumulative and variation LAI, with a minimum DHI correlation of 0.11.

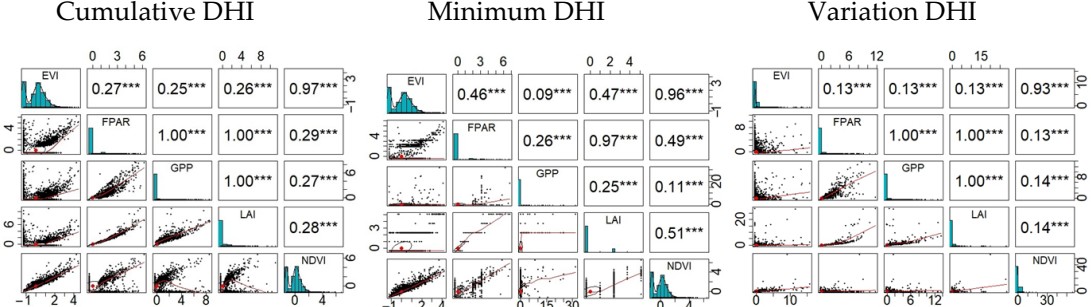

**Figure 3.** Spearman rank correlations and scatter plots among the three DHIs derived from various MODIS products. Significant relations: *** $p < 0.001$.

Due to the high correlation between FPAR-based DHIs and species richness, we selected FPAR-DHIs for the subsequent work analysis (Tables 3 and S1–S4). Most of the correlations between the FPAR-based DHIs and the climate dataset were weak (Figure 4), indicating that the DHIs possibly provide various information. The highest Spearman rank correlation value with the energy factor was −0.13 (BIO4, Temperature Seasonality) during the cumulative DHIs. Regarding the water factor, the highest Spearman rank correlation measured 0.07 (BIO13, the wettest month).

**Table 3.** Coefficient of determination between the individual FPAR-based DHIs of the four main taxa. Significant relations: ** $p < 0.01$; *** $p < 0.001$.

| | Cumulative DHI R² Adj | RMSE | Minimum DHI R² Adj | RMSE | Variation DHI R² Adj | RMSE | All DHIs R² Adj | RMSE |
|---|---|---|---|---|---|---|---|---|
| Mammals | 0.12 *** | 7.03 | 0.03 *** | 7.38 | 0.12 *** | 7.04 | 0.14 *** | 6.98 |
| Birds | 0.20 *** | 37.10 | 0.04 *** | 40.59 | 0.19 *** | 37.42 | 0.25 *** | 36.73 |
| Breeding birds | 0.20 *** | 34.95 | 0.05 *** | 38.17 | 0.18 *** | 35.40 | 0.24 *** | 34.66 |
| Non-breeding birds | 0.00 *** | 18.97 | 0.00 ** | 19.01 | 0.00 *** | 18.98 | 0.01 *** | 18.97 |

### 3.2. Species Richness and Productivity Hypothesis

The high density of a species is mainly spread in the marginal regions of the Kunlun–Pamir Plateau, characterized by relatively small areas (Figure 5). The dense distribution of species is primarily detected in the frontier areas of the Kunlun–Pamir Plateau, with relatively small but insignificant differences in species' abundance in the central region (Figure 5). The species richness correlated positively with the cumulative DHI based on the FPAR for all four categories, with the Spearman rank correlation coefficients amounting to 0.36, 0.39, 0.39 and 0.08 for mammals, birds, breeding birds and non-breeding birds, respectively. However, there was also a considerable dispersion noticeable. The minimum DHI based on the FPAR predicted that the species richness was higher when the minimum DHI based on the FPAR was larger. In this case, the positive correlation coefficients were 0.19, 0.22, 0.22 and 0.04 for mammals, birds, breeding birds and non-breeding birds, respectively. The scatter plots of the minimum DHI based on the FPAR versus the species richness for the four taxa were similar to those of the cumulative DHI based on the FPAR and variation DHI based on the FPAR. The variation DHI based on the FPAR predicted that

the species richness would be higher when there was less variability. However, because most parts of the Kunlun–Pamir Plateau region are bare land and desert steppe, the species richness was higher when the environmental conditions were more variable. Indeed, the variability in the Kunlun–Pamir Plateau region correlated positively with the species richness for all four taxa, with positive correlation coefficients of 0.36, 0.4, 0.4 and 0.08 for mammals, birds, breeding birds and non-breeding birds, respectively. This implies that, for all four taxa, high values of the variability DHI based on the FPAR indicate a high species richness variation.

Furthermore, the three DHIs based on the FPAR complement each other (Figure 6 and Table 3). A hierarchical partitioning analysis of the multiple regression models showed that all three DHIs based on the FPAR contributed considerably to the explanation of the overall variability in species richness. The cumulative DHI based on the FPAR was the most critical variable in each regression model, followed by the varDHI based on the FPAR and MinDHI based on the FPAR. According to the regression models, birds demonstrate the most explanatory power, and non-breeding birds exhibit the least explanatory power among these four categories (Figure 6).

### 3.3. Species Richness and the Water–Energy Dynamics Hypothesis

The climatic factors affected the distribution of the species richness in the Kunlun–Pamir Plateau region (Table 4). Among the bioclimatic variables, BIO12, -14, -17 and -19 denoted the highest rank correlation coefficients in species richness. Although the differences between the four variables were not statistically significant, BIO17 demonstrated the highest rank correlation coefficient regarding the species richness. Spearman's rank correlation coefficients with the BIO17 Precipitation of the Driest Quarter measured 0.61, 0.73, 0.72 and 0.14 for mammals, birds, breeding birds and non-breeding birds, respectively. The higher BIO17 corresponded to a larger species richness. Among the four taxa, the birds and breeding birds demonstrated more substantial Spearman rank correlation coefficients with climate factors as compared to the mammals and non-breeding birds.

We noticed that all BIOs (12, 14, 17 and 19) contributed more to the overall richness variability in the multiple linear regression. A hierarchical partitioning analysis of the multiple regression models showed that the four bioclimatic variables' influence complemented the definition of the overall richness variability. BIO17 identified as being the most important for mammals and birds, while BIO14 was the most striking for the breeding birds and non-breeding birds. The minimum DHI was the least effective for all models. Among the significant taxa, the breeding bird's regression model explained most of the variation in species richness (Figure 7 and Table 4).

### 3.4. The Combined Effect of the Productivity Hypothesis and Water–Energy Dynamics Hypothesis on Species Richness

The productivity hypothesis and water–energy dynamics hypothesis explained the distribution of species richness in the Kunlun–Pamir Plateau from a different aspect (Figure 8). Combining the two hypotheses could help to fully understand the species richness distribution in the Kunlun–Pamir Plateau. Our analysis of the combination of two hypotheses demonstrated that the explanatory power of the multiple regression models on species richness improved when compared to the single regression models based on each hypothesis separately, except for the non-breeding birds (Table 5). In particular, the model improved from 0.24 to 0.32 for mammals, 0.54 to 0.66 for birds and 0.54 to 0.65 for breeding birds. The multiple regression model's hierarchical partitioning analysis revealed that all seven influences played a key role when explaining the overall variability in species richness. BIO17 identified the most significant species richness regressions for the mammals, birds and breeding birds, while BIO12 was the most important for the non-breeding birds. The minimum DHI based on the FPAR was the least crucial for all models. The regression model for birds mainly justified the species richness variation among the four taxa (Figure 9).

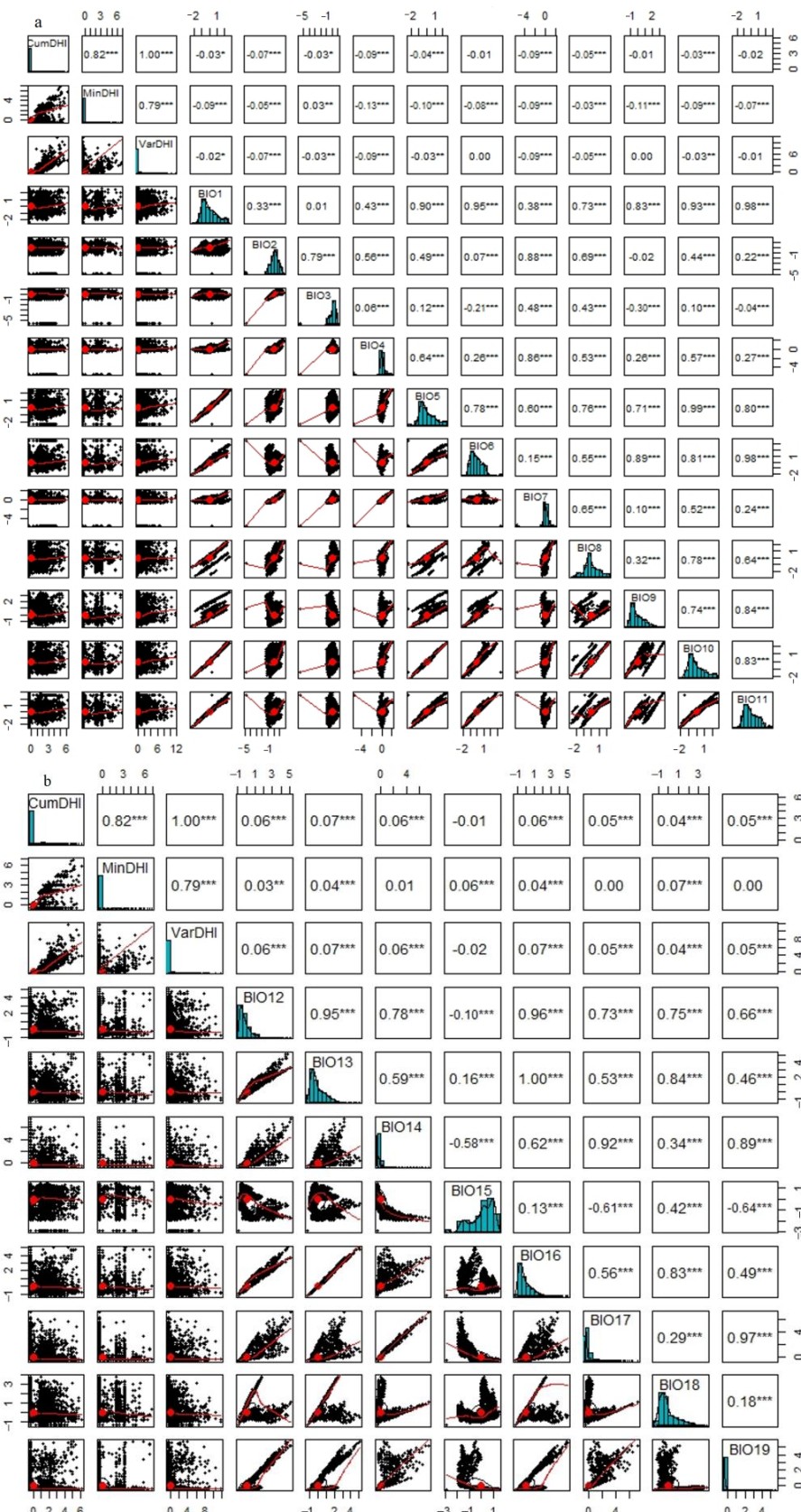

**Figure 4.** Pairs of the Spearman correlations of all MODIS FPAR-based DHIs versus the environmental variables grouped by (**a**) DHIs and temperature variables from Bioclim and (**b**) DHIs and precipitation values from Bioclim. Significant relations: * $p < 0.05$; ** $p < 0.01$; *** $p < 0.001$.

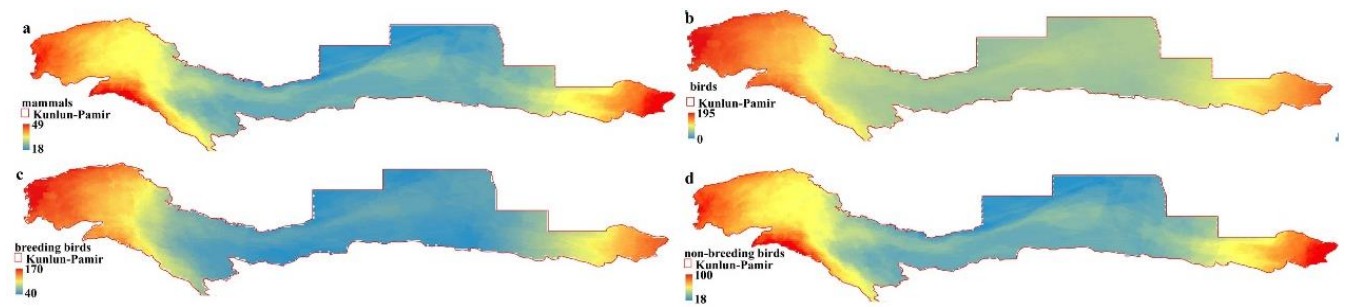

**Figure 5.** Distribution of species richness of (**a**) mammals, (**b**) birds, (**c**) breeding birds and (**d**) non-breeding birds.

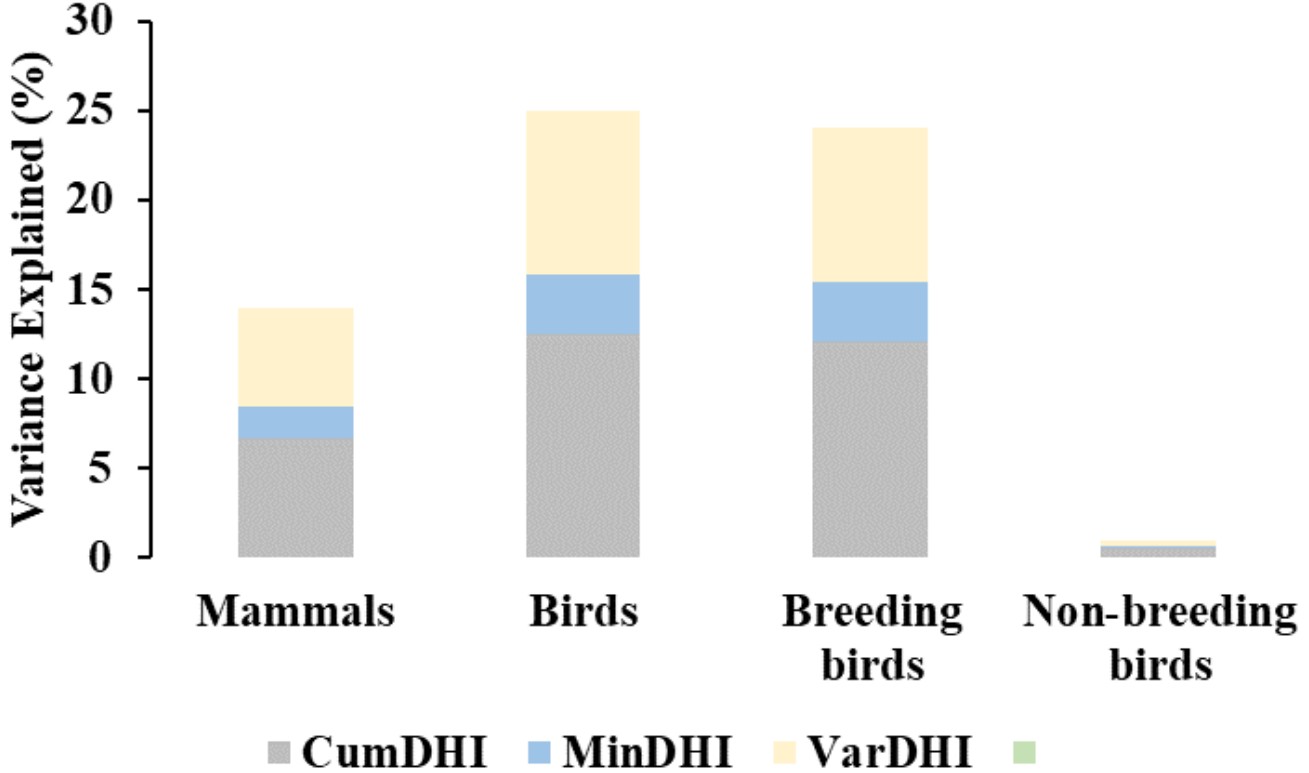

**Figure 6.** The relative importance of the three FPAR-based DHIs to the four main taxa.

**Table 4.** Coefficient of determination between the Bioclim variables (BIO12, BIO14, BIO17 and BIO19) and four main taxa. Significant relations: * $p < 0.05$; *** $p < 0.001$.

| | BIO12 R² Adj | RMSE | BIO14 R² Adj | RMSE | BIO17 R² Adj | RMSE | BIO19 R² Adj | RMSE | The Four BIOs R² Adj | RMSE |
|---|---|---|---|---|---|---|---|---|---|---|
| Mammals | 0.24 *** | 6.54 | 0.20 *** | 6.70 | 0.23 *** | 6.59 | 0.21 *** | 6.65 | 0.29 *** | 6.32 |
| Birds | 0.54 *** | 28.12 | 0.49 *** | 29.60 | 0.53 *** | 28.48 | 0.52 *** | 28.91 | 0.61 *** | 25.82 |
| Breeding birds | 0.54 *** | 26.55 | 0.49 *** | 27.92 | 0.52 *** | 27.07 | 0.50 *** | 18.85 | 0.60 *** | 24.69 |
| Non-breeding birds | 0.02 *** | 18.84 | 0.02 *** | 18.86 | 0.02 *** | 18.85 | 0.02 * | 18.85 | 0.02 *** | 18.82 |

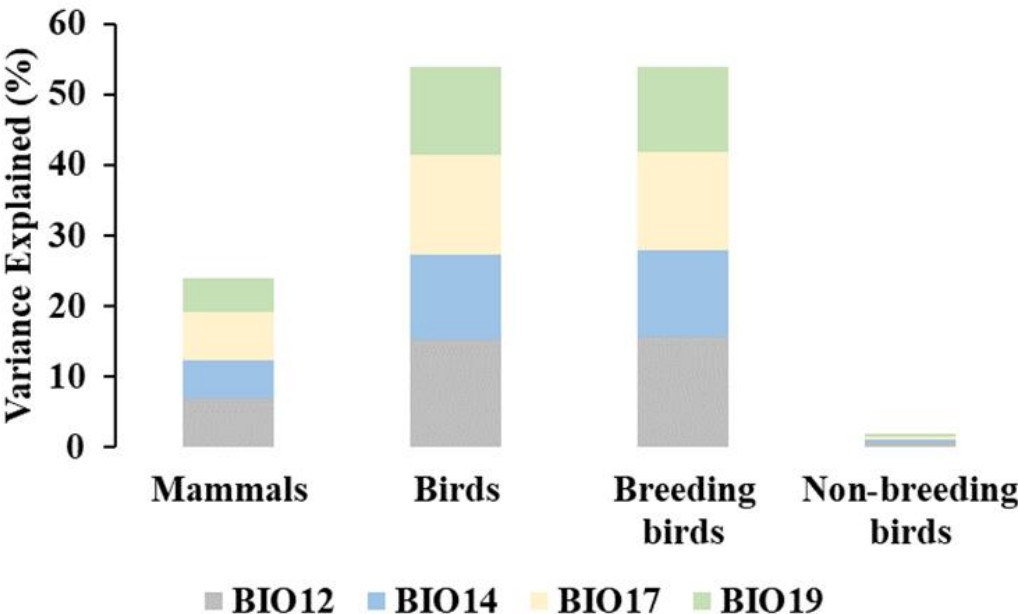

**Figure 7.** The relative importance of the four Bioclim variables and the four main taxa.

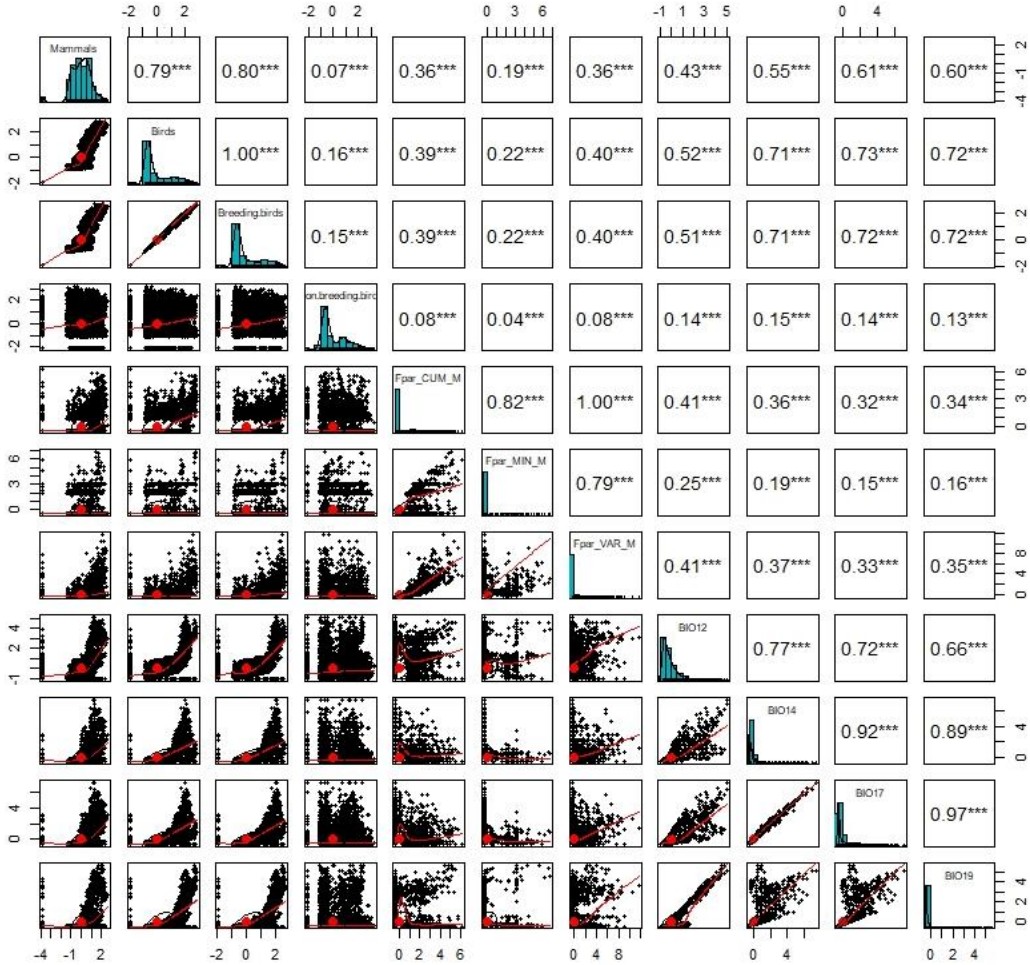

**Figure 8.** Pairs of the Spearman correlations of all MODIS FPAR-based DHIs and the four Bioclim variables (BIO12, BIO14, BIO17 and BIO19) versus the species richness of the four taxa (mammals, birds, breeding birds and non-breeding birds). Significant relations: *** $p < 0.001$.

**Table 5.** Coefficient of determination between the four Bioclim variables (BIO12, BIO14, BIO17 and BIO19), as well as the FPAR-based DHIs to the three significant relations: *** $p < 0.001$.

| | Four BIOs and Three FPAR-Based DHIs $R^2$ Adj | RMSE |
|---|---|---|
| Mammals | 0.32 *** | 6.18 |
| Birds | 0.66 *** | 24.34 |
| Breeding birds | 0.65 *** | 23.28 |
| Non-breeding birds | 0.02 *** | 18.82 |

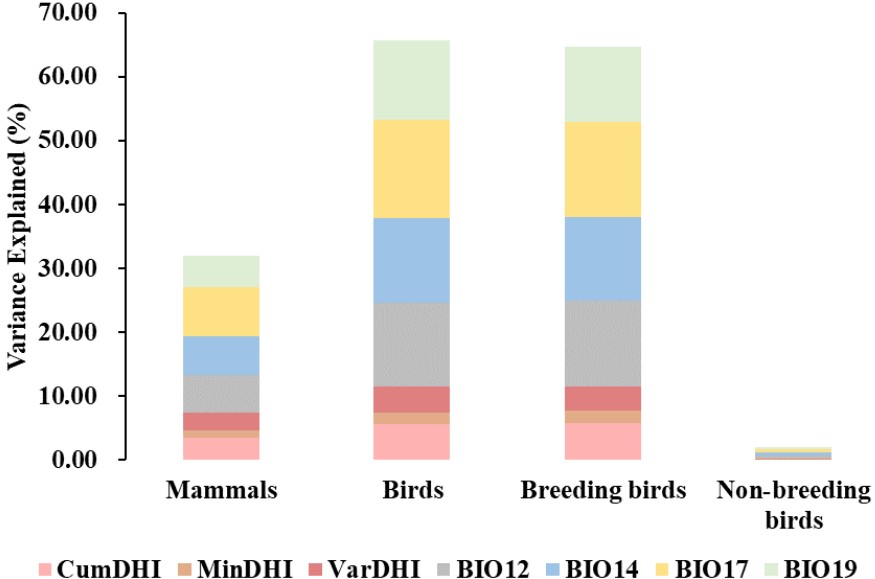

**Figure 9.** The relative importance of (the) three FPAR-based DHIs, as well as four Bioclim variables to the four taxa.

## 4. Discussion

### 4.1. The Application of the Productivity Hypothesis

Revealing the species richness patterns is essential for the biodiversity, especially for the conservation of endangered species in the Kunlun–Pamir Plateau area. However, only a few studies have focused on the biodiversity in this region. We constructed five DHIs based on the MODIS NDVI, EVI, FPAR, LAI and GPP data, and we assessed their performance in the species richness models on the Kunlun–Pamir Plateau. The findings revealed that the three DHIs (CumDHI, MinDHI and VarDHI) correlated weakly with the species richness, suggesting that the concept of the productivity hypothesis does not apply well to the desert plateau locations. Among these four taxa, the explanatory power of the FPAR-based multiple regression models (for DHIs) was revealed to be weaker when compared to the application of the DHIs in other regions (Table 3). However, the correlations appeared the most vital for birds and least worthwhile for the non-breeding birds. Our findings were consistent with the results of other studies [22]. According to earlier research [27,28], the FPAR captures more subtle changes in the grassland photosynthetic activity; therefore, derived DHIs perform best regarding the grassland breeding species, and our results support this. In addition, contrary to other studies, the land cover in the Kunlun–Pamir Plateau region is dominated by bare land and grasslands, in which the species richness correlated positively with the VarDHI based on the FPAR, making the results more reasonable. However, the DHIs based on the FPAR in this study area did not indicate an obvious spatial pattern relative to the large-scale studies. Moreover, the correlations between the species richness and the DHIs based on the FPAR were rather small, demonstrating that the applicability of the DHIs based on the FPAR still needs further testing in relatively sparse vegetated mountainous areas.

*4.2. The Application of the Water–Energy Dynamic Hypothesis*

Understanding the factors affecting the species' distribution and the diversity patterns is essential to conserve the regional species richness [11,42]. The water–energy dynamics hypothesis interpreted the main influencing factors affecting the species in the Kunlun–Pamir Plateau region regarding energy and moisture. The results showed that the moisture significantly influenced the species richness more than energy in the Kunlun–Pamir Plateau area. The correlations between the variables related to the moisture factor, and the mammals, birds, breeding birds and non-breeding birds were 0.29, 0.61 and 0.60, respectively (Table 4). The moisture had a relatively important influence on the Kunlun–Pamir Plateau area, which confirmed O'Brien's conclusion regarding the main influencing factors on the species richness distribution patterns in arid regions [43]. The water–energy dynamics hypothesis was applicable to explain the species richness distribution patterns. Moisture is the main factor affecting the species richness in the Kunlun–Pamir Plateau region due to the continental alpine climate with sufficient sunshine and low rainfall [44]. As the plateau continues to warm, the snowpack will decrease or fluctuate, which may significantly impact the species richness [45]. Therefore, in the Kunlun–Pamir Plateau region, further research on how climate change effects biodiversity will be essential.

*4.3. Limitations*

In this study, we investigated the factors that affect the species richness in the Kunlun–Pamir Plateau region only from the view of two perspectives: the productivity hypothesis and the water–energy dynamics hypothesis. However, an increasing number of studies have already shown that the mechanisms underlying the geographic patterns of species richness are complex and cannot be explained by only one hypothesis [9,11]. Thus, the productivity and water–energy dynamics hypotheses were joined in this work, and various regression models were employed to improve the model's interpretation of species richness. However, the species richness change was also influenced by land use change, invasive alien species and other elements [2,46–49]. More factors still need to be considered among the factors impacting the species richness patterns, such as the topographic factors [50], human disturbance [47] and spatial correlation [8,51]. Future studies should include more causal factors for species richness in order to reveal the mechanism that maintains the species richness distribution pattern of the Kunlun–Pamir Plateau more thoroughly.

**5. Conclusions**

We analyzed the species richness pattern in the Kunlun–Pamir Plateau region, taking into account both the productivity hypothesis and the water–energy dynamics hypothesis.

From the aspect of the productivity hypothesis, we investigated the correlation between the dynamic habitat indices (DHIs) and the species richness in the Kunlun–Pamir Plateau region. The calculated DHIs by means of five MODIS products correlated less with the mammal, bird, breeding bird and non-breeding bird richness in the Kunlun–Pamir Plateau region. The DHIs based on the FPAR data correlated slightly higher with the mammal, bird, breeding bird and non-breeding bird richness than those based on the other four MODIS product DHIs. Although the cumulative DHIs based on the FPAR conformed more with the richness of the four taxa species compared to the variation based on the FPAR and minimum DHIs based on the FPAR, the multiple regression models for the three variables associated better with the richness of the four species. However, the DHI based on the FPAR correlation coefficients with an abundance of mammals, birds, breeding birds and non-breeding birds were still low, namely 0.24, 0.25, 0.24 and 0.01, respectively. The birds regression model clarified most of the variations in species richness during the four taxa.

Regarding the water–energy dynamics hypothesis, the precipitation-related climatic factors dominate the distribution of species richness on the Kunlun–Pamir Plateau; among those, BIO12 had the highest correlations with mammals, birds, breeding birds and non-breeding birds, amounting to 0.24, 0.54, 0.54 and 0.02, respectively, followed by BIO17,

-19 and -14. Multiple linear regression models (based on four climate factors) raised the correlations between these factors with the mammals, birds, breeding birds and non-breeding birds (to 0.29). Furthermore, we noticed that the correlation between the species richness and the climatic factors was noticeably higher in the Kunlun–Pamir Plateau than the MODIS-based DHIs. The correlations between the DHIs based on the FPAR and the climate factors increased to 0.32, 0.66, 0.65 and 0.02, respectively, for the multivariate linear regression models. The regression model for birds also explained the majority of the variations in species richness for the four taxa. The water–energy dynamic hypothesis was more appropriate to the Kunlun–Pamir Plateau region.

In general, by examining the DHIs and the species richness in specific areas, we suggest a reconstruction of the DHIs appropriate for the region, especially for locations with relatively small spatial scales. In the context of climate change, our results once again highlight the significance of monitoring climate fluctuations in the Kunlun–Pamir region for biodiversity conservation. This will make it easier to evaluate the spatial distribution of species richness and provide an accurate assessment of that distribution.

**Supplementary Materials:** The following supporting information can be downloaded at: https://www.mdpi.com/article/10.3390/rs14246187/s1: Table S1: Coefficient of determination between the individual EVI -based DHIs of the four main taxa; Table S2: Coefficient of determination between the individual GPP -based DHIs of the four main taxa; Table S3: Coefficient of determination between the individual LAI -based DHIs of the four main taxa; Table S4: Coefficient of determination between the individual NDVI -based DHIs of the four main taxa; Table S5: Coefficient of determination between the individual NDVI -based DHIs of the four main taxa.

**Author Contributions:** Conceptualization, X.H. and A.B.; methodology, X.H. and A.B.; data curation and software, J.Z.; validation, X.H. and T.Y.; formal analysis, X.H. and G.Z; resources, X.H.; data curation, X.H.; writing—original draft preparation, X.H. and G.Z.; writing—review and editing, A.B., Y.Y., T.W., V.N., P.D.M. and T.V.d.V.; visualization, X.H.; supervision, A.B.; project administration, A.B. and funding acquisition, A.B. All authors have read and agreed to the published version of the manuscript.

**Funding:** This research was supported by the Strategic Priority Research Program of the Chinese Academy of Sciences (Grant No. XDA20030101), the Qinghai Province Kunlun talents—leading talents project (2020-LCJ-02).

**Conflicts of Interest:** The authors declare no conflict of interest.

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
