# Peer review of "Precipitation Dominates the Distribution of Species Richness on the Kunlun–Pamir Plateau"

_remotesensing, doi:10.3390/rs14246187_

Round 1

Reviewer 1 Report

In this manuscript the authors examine the Kunlun-Pamir Plateau, a biodiversity reserve of importance, and investigate the species richness distribution patterns and their correlation with the productivity and water-energy hypotheses. The productivity hypothesis is assess using MODIS-derived indices to estimate three Dynamic Habitat Indices (DHIs): 1) CumDHI, 2) MinDHI, 3) VarDHI. The water-energy hypothesis is assessed based on a series of bioclimatic variables. Literature review presented is sufficient and covers the required aspects. “Materials and methods” sections is clearly understood, and the proposed methodology is elaborated. Results presented are explained and supported. Writing, grammar, and syntax are on a good level, although some minor errors can be observed. In general, the manuscript is a little difficult to follow at certain points and some writing revisions would be advised to better promote clarity and coherence. I conclude that the manuscript is accepted after some revisions.

General comments on the manuscript:

Page 2 – line 90

Isn’t the goal to explain species richness distribution using DHIs and bioclimatic factors? This paragraph needs to be adjusted to better reflect the objectives of this research. Conclusions should then follow the changes that took place in this paragraph.

Page 3 – line 109

Incorrect reference to a figure that is after 3 pages and the data and not the appropriate.

Page 3

Is there any other information that can be described for the species richness and bioclimatic data? e.g how often are these data updated, etc…?

Page 4:

The equations have been pasted as images most likely and they appear pixelated. Please change them in order for the material to be appear better.

Page 4 – line 132 “where VarDHI stands for the variation of the production values in one year”

In the equation there isn’t any mention to the VarDHI but the placement of the sentence suggests otherwise. Please correct.

Page 4 – line 138-140.

Text appears as different size from the rest.

Page 4 line 147 “random sample of 10,000 of the 1 km DHIs” and page 5 line 155 “based on a sample of 10,000 of the 10km 155 resampled DHIs and Bioclim variables.”

Perhaps both should be at the same resolution of 1km?

Page 6

It is not clearly evident why the DHI correlations from the same MODIS vegetation dataset are assessed? What is the importance of this and how this contributes to the objectives stated in the beginning. The same applies to the correlation among the three DHIs derived from various MODIS products.

Page 7

Why were the DHIs derived from FPAR chosen from the DHIs derived from other products?

Reviewer 2 Report

Thank you for providing me with this opportunity to review the entitled manuscript " Precipitation dominates the distribution of species richness on the Kunlun-Pamir Plateau". It is quite interesting to work and the authors have made a good attempt to evaluate the water-energy dynamics’ hypotheses to investigate the distribution pattern of the species richness (and its determinants) in the Kunlun-Pamir Plateau. Research design is appropriate and discussed with recent literature. The manuscript is relatively completed. I, therefore, suggest accepting it in its present form. Authors need to check the spelling and grammar mistakes. 

Reviewer 3 Report

Dear Authors:

Many of the figures are, essentially, unreadable: the font size is simply too small to read the contents of the figures.

 Line 209—you reference “Figure 9” but I believe this is not correct.

 Vertical axis in Fig. 6 should not extent to “-4”—it should have “0” as a minimum value.

 Table 3 and all others like it: the table title indicates “regression coefficients” but the contents of the table are R2 values—an R2 value is not a regression coefficient but rather a coefficient of determination.

 Table 4 presents an R2 = 0 as significant…taken at face value (and realizing that there  is of course some rounding to 2 decimals places), you are saying that an estimated R2 that rounds to 0.00 is significant—do you really want to say that?

 The single variable analyses used Spearman correlations, but apparently the multiple regression analysis is using an R2 which is a Pearson-based statistic.  Justify why you are using a rank-based correlation analysis but not a rank-based multiple regression?

 The interpretation of the results in lines 270ff are self-evident: assuming that the numbers provided are R2 values, of course a multiple regression model will have a higher R2 value as the number of independent variables increase (assuming the very unlikely event that an added variable provides precisely 0 additional sum of squares).  What needs to be done here is a formal test that these models are statistically better, and, of course, with your sample sizes, they will be significantly better. I would strongly urge the authors to go through formal model selection process and well as formal model validation of the final models (perhaps computing an R2 based on a prediction error sum of squares…something to provide a meaningful interpretation given that the sample size is so large).

Taken at face value, main results of the paper seem to be that “species richness in this area is positively related to rainfall” but that scarcely (~ < 0.6) more than half the variation in richness is explained by the water-energy hypothesis.  This is a reasonable result. It also seems rather self-evident…and somewhat limited in its usefulness.  Formal model selection might well show that the water-energy hypothesis is more applicable than the productivity hypothesis, but what the practical implications (if any) are from this result is not discussed by the authors.

Reviewer 4 Report

In this paper, the authors analyzed the species richness pattern in the Kunlun-Pamir Plateau region, considering both the productivity and the water-energy dynamics hypotheses. This research is well written in standard language. However, the biggest problem of this work is the lack of innovation. The authors only applied the off-the-shelf index (DHIs) to assess the correlation of different factors with the species richness, which made no further extension and improvement. In this case, I decide to reject this paper. Besides that, there are some minor problems needed to be concerned.

Point 1: line 288: The term “non-breeding birds” appeared twice.

Point 2: For the expression “The calculated DHIs by means of five MODIS products) correlated less…”, the left bracket is missing here.

Reviewer 5 Report

Please describe in a paragraph why specifically did you choose to study the Kunlun-Pamir Plateau. 

Round 2

Reviewer 4 Report

By considering both the productivity and water-energy dynamics hypotheses, the authors studied the distribution of species richness in the Kunlun-Pamir Plateau using publicly available datasets. Several statistical indices, including CumDHI, MinDHI, and VarDHI (from NDVI, EVI, FPAR, LAI, and GPP, respectively), were generated for the analytical method. What could be drawn from the data collected from the Kunlun-Pamir Plateau? The conclusion was drawn using the mentioned DHIs in this work. These indices were adopted in the research on the distribution of species richness or similar objectives on other terrain environments and proven to be suitable, but were the calculated statistical characteristics presented the most suitable indices for revealing the distribution of species richness? Regarding the statistical analysis, any other correlation analytical methods are appropriate? Comparison using different methods is usually needed to avoid the potential special data-dependent modeling mechanism, if possible.
